# *De novo* assembly, annotation, marker discovery, and genetic diversity of the *Stipa breviflora* Griseb. (Poaceae) response to grazing

**Dongqing Yan[1]ʘ, Jing Ren[1]ʘ, Jiamei Liu[1], Yu Ding[1], Jianming Niu[1,2,3]***

**1** School of Ecology and Environment, Inner Mongolia University, Hohhot, People's Republic of China,
**2** Inner Mongolia Key Laboratory of Grassland Ecology and the Candidate State Key Laboratory of Ministry of Science and Technology, Hohhot, People's Republic of China, **3** Ministry of Education Key Laboratory of Ecology and Resource Use of the Mongolian Plateau, Hohhot, People's Republic of China

ʘ These authors contributed equally to this work.
* jmniu2005@163.com

**Data Availability Statement:** All transcriptome data have been deposited in the NCBI Sequence Read Archive (SRA) with accession number as

## Abstract

Grassland is one of the most widely-distributed ecosystems on Earth and provides a variety of ecosystem services. Grasslands, however, currently suffer from severe degradation induced by human activities, overgrazing pressure and climate change. In the present study, we explored the transcriptome response of *Stipa breviflora*, a dominant species in the desert steppe, to grazing through transcriptome sequencing, the development of simple sequence repeat (SSR) markers, and analysis of genetic diversity. *De novo* assembly produced 111,018 unigenes, of which 88,164 (79.41%) unigenes were annotated. A total of 686 unigenes showed significantly different expression under grazing, including 304 and 382 that were upregulated and downregulated, respectively. These differentially expressed genes (DEGs) were significantly enriched in the "alpha-linolenic acid metabolism" and "plant-pathogen interaction" pathways. Based on transcriptome sequencing data, we developed eight SSR molecular markers and investigated the genetic diversity of *S. breviflora* in grazed and ungrazed sites. We found that a relatively high level of *S. breviflora* genetic diversity occurred under grazing. The findings of genes that improve resistance to grazing are helpful for the restoration, conservation, and management of desert steppe.

## Introduction

Chinese grasslands are diverse and constitute the third largest grassland area worldwide, covering 41.7% of the country's territory and stretching from northern China to the Qinghai-Tibetan Plateau [1–3]. These grasslands are currently in danger of degradation owing to climate change and anthropogenic activities, and over 33% of the degradation is due to overgrazing [4, 5]. Long-term overgrazing impacts species richness, community biomass, and soil quality [6–8]. However, plants can adapt to grazing pressure by developing certain characteristics, such as small sizes, fast growth, short life-spans, and low palatability, to avoid livestock

SRR9903413, SRR9903414, SRR9903415, SRR9903416, SRR9903417, and SRR9903418.

**Funding:** This study was supported by Natural Science Foundation of China (NSFC) (31460154, 31860106), and Major Science and Technology Projects of Inner Mongolia Autonomous Region (2019ZD008). JN received the above three awards. The funders had no role in study design, data collection and analysis, decision to publish, or preparation of the manuscript.

**Competing interests:** The authors have declared that no competing interests exist.

feeding and increase the survival rate [9]. *Stipa* species (Poaceae) are the most conspicuous grassland plants distributed throughout the Eurasian Steppe and are established as zonal vegetation in the Inner Mongolia grassland [10–13]. *S. breviflora* Griseb. is one of the dominant species and valuable forage resource in the desert steppe and is characterized by palatability, rich nutrient content, and early greening. Additionally, this species has strong ecological adaptability to diverse environments and can be a codominant species with *S. bungeana* and *S. krylovii* in warm-temperate steppe and typical steppe, respectively [14–19]. Although *S. breviflora* is vulnerable to climate change and grazing, it still has great potential for expansion and evolution via hybridization with other feather grasses [13, 17, 20]. Thus, there is a need for further elucidation of the adaptation of *S. breviflora* to grazing.

Genetic diversity is fundamental to biodiversity [21]. Climate factors, such as temperature and precipitation, and human activities impact genetic diversity, resulting in changes in genetic differentiation [22, 23]. Studies have shown that temperature is the key factor in determining the genetic differentiation of *S. breviflora* [14]. Anthropogenic activity, such as grazing, can increase, decrease, or have no significant effects on the genetic diversity of plants. In the genus *Stipa*, moderated grazing was found to promote the genetic diversity of *S. grandis* and *S. krylovii* [24]. Similar results were detected in the study by Shan et al. [25], and the genetic diversity of *S. grandis* under light grazing conditions was higher than that under no grazing. Other plants, such as *Elymus nutans*, had higher intrapopulation genetic diversity in three grazed areas than in ungrazed areas [26]. However, a negative relationship between grazing and genetic diversity was also observed. For example, *Artemisia frigida* displayed restricted gene flow and decreased genetic diversity under grazing [27]. In some cases, grazing had no significant effect on plant genetic diversity. Smith et al. [28] investigated the fine-scale spatial genetic structure in *Bouteloua curtipendula var. caespitosa*, *Bouteloua gracilis* and *Poa ligularis* and observed no differences in average gene diversity between populations of each species. A study on *Festuca idahoensis* revealed that grazing did not significantly alter their genetic diversity [29]. In summary, although plants respond to climate and human activities by changing their genetic diversity, the molecular mechanisms driving these changes remain poorly understood.

The next-generation sequencing (NGS) technology have provided an efficient way to generate an abundance of genomic data. Compared with traditional low-throughput EST (Expressed Sequence Tag) sequencing by Sanger technology, RNA-sequencing (RNA-seq) offers more transcripts for both marker development and gene discovery, helping us to explore the molecular mechanism of genetic diversity [30]. Using transcriptome sequencing, Ren et al. [31] developed 21 microsatellite markers of *S. breviflora* and tested the polymorphism in six related species. Klichowska et al. [32] mined 10 microsatellite markers of *S. pennata* using Illumina high-throughput. Transcriptome analysis has also been used to uncover the adaptive mechanisms of plants subjected to grazing. For *Leymus chinensis*, genes related to the systemic synthesis of jasmonate are activated during grazing [33]. Wang et al. [34] conducted *de novo* assembly to compare the transcriptomes of two alfalfa varieties that are tolerant and intolerant to grazing, and identified 21 differentially expressed responsive pathways. For *S. grandis*, individuals subjected to grazing develop smaller size than those not subjected to grazing, and the expression of wound-, drought-, and defense-related genes is altered [35]. However, the response of *S. breviflora* to grazing at the transcriptome level and the changes in genetic diversity remain unclear.

Considering the crucial position of *S. breviflora* in desert steppe and to meet the needs for exploring the adaptation mechanism of grasses to grazing, we took this species as an example to (1) investigate the transcriptome response to grazing; (2) develop simple sequence repeat (SSR) molecular markers based on transcriptome sequencing data; and (3) explore the impact of grazing on the genetic diversity. The present study provides insight into the adaptation mechanisms of plants under grazing.

## Materials and methods

### Collecting plant materials

The study area is in Wuchuan, Inner Mongolia Autonomous Region, China (41˚47′-41˚23′N, 110˚31′-111˚53′E) at an elevation of 1500–2000 m above sea level. This area belongs to the monsoon climate of medium latitudes. The mean annual precipitation and temperature are 354.1 mm and 2.6˚C, respectively. We observed three ungrazed sites (codes 1, 2, and 3) and three grazed sites (codes 4, 5, and 6) (Table 1). The ungrazed sites have been fenced to prevent grazing for many years. In contrast, the grazed sites have been subjected to long-term heavy grazing. The area of each site is more than 1 hm$^2$. These sites are distributed adjacently and are approximately around 2–10 km apart, indicating that they are under similar backgrounds of temperature and precipitation. The study was carried out in public land. No specific permissions are required for collecting samples, and no endangered or protected species are involved in study area.

In June 2016, we collected 33 samples at each site, with 3 samples for RNA-seq and 30 samples for genetic diversity analysis. For RNA-seq, we chose healthy and intact individuals in ungrazed sites, and chose individuals in obviously grazed patches in grazed sites. Samples for RNA-seq were rinsed with RNase-free dd water, mixed with equal amounts, stored in dry ice, and sent to the Beijing Genomics Institute (BGI, Shenzhen, China) for total RNA extraction and high-throughput sequencing. For genetic diversity analysis, we chose the healthy and intact individuals that were at least 10 m apart in both grazed and ungrazed populations. All samples were immediately frozen in liquid nitrogen and brought back to the Genetics Lab and stored at -80˚C for genetic diversity analysis.

### RNA preparation and cDNA library construction

The total RNA of each sample was extracted using Trizol reagent (Invitrogen, Carlsbad, CA, USA) according to the manufacturer's protocol. RNA integrity was examined using an Agilent 2100 Bioanalyzer (Agilent Technologies, Santa Clara, CA, USA), and poly (A) mRNA was isolated using Oligo (dT) beads. The mRNA fragments were used as templates for reverse transcription to obtain cDNA. After fragment ends were repaired, polymerase chain reaction (PCR) amplification was performed, followed by poly (A) ligation and fragment length selection to construct a cDNA library. To obtain a high-quality library, an Agilent 2100 Bioanalyzer and ABI StepOnePlus real-time PCR system were used for quality control analysis.

### RNA-seq and *de novo* assembly

The libraries were sequenced using the Illumina HiSeq 4000 platform (Illumina, USA). The raw reads generated by sequencing were filtered by eliminating adaptor sequences and low-quality reads and then assembled by the Trinity program with the default parameters [36]. The longest transcripts of each gene were chosen as unigenes.

**Table 1. Information of the observed sites of *S. breviflora*.**

| Code | Position | Altitude(m) | Utilization | Plant community |
|------|----------|-------------|-------------|-----------------|
| 1 | 111.21˚E, 41.13˚N | 1629 | fenced without grazing | *S. breviflora* + *S. krylovii* |
| 2 | 111.26˚E, 41.19˚N | 1680 | fenced without grazing | *S. breviflora* + *S. krylovii* + *Thymus mongolicus* |
| 3 | 111.19˚E, 41.11˚N | 1668 | fenced without grazing | *S. breviflora* + *S. krylovii* |
| 4 | 111.19˚E, 41.11˚N | 1647 | sheep grazing | *S. breviflora* + *S. krylovii* + *Thymus mongolicus* |
| 5 | 111.29˚E, 41.25˚N | 1614 | sheep grazing | *S. breviflora* + *Agropyron cristatum* |
| 6 | 111.31˚E, 41.29˚N | 1572 | sheep grazing | *S. breviflora* + *S. krylovii* + *Cleistogenes squarrosa* |

## Unigene annotation

Unigenes were aligned using the basic local search tool (BLAST) [37] against databases, including the nucleotide collection (Nt, https://www.ncbi.nlm.nih.gov/), National Center for Biotechnology Information (NCBI) non-redundant protein (Nr, https://www.ncbi.nlm.nih.gov/), Cluster of Orthologous Groups (COG, http://www.ncbi.nlm.nih.gov/COG), Kyoto Encyclopedia of Genes and Genomes (KEGG, http://www.genome.jp/kegg) and Swiss-Prot (http://ftp.ebi.ac.uk/pub/databases/swissprot) databases. Blast2GO [38] was used to annotate the unigenes against gene ontology (GO, http://geneontology.org) based on the results of the Nr annotation.

InterProScan5 [39] was used for InterPro (http://www.ebi.ac.uk/interpro) annotation. According to the functional annotation results, the best-matched sequences were defined as complete coding sequences (CDS) of the corresponding unigenes and saved in FASTA format, with the following order of priority: Nr, SwissProt, KEGG, and COG. Moreover, the predicted CDS were used as a model to predict the unigenes without matches in the abovementioned database through modeling with ESTScan [40].

## Differentially Expressed Gene (DEG) identification

The expression levels of each gene were calculated by RSEM [41]. Differential expression analysis of genes in grazed and ungrazed samples was performed using the NOIseq approach [42]. We set fold change $\geq$ 2.00 and probability $\geq$ 0.8 as thresholds to identify the genes that were expressed at a significantly different level.

## Developing grazing-related SSRs

MISA [43] and Primer3 [44] were used to detect SSRs from the transcriptome and design primers, respectively. We selected 21 SSR molecular markers derived from DEGs to screen grazing-related SSRs and then determined the optimal annealing temperature by PCR amplification (S1 Table). Total genomic DNA was extracted using the TIANGEN plant genomic DNA kit (Tiangen Biotech, Beijing, China) for all samples following the manufacturer's instructions. The quality and quantity of DNA were determined by using a NanoDrop 2000 spectrophotometer (Thermo Scientific, Wilmington, DE, USA). The PCR amplification reaction mixture (25 μL) contained 1 μL template DNA (30–40 ng μL–1), 0.5 μL (10 pM) of each primer, 12.5 μL Premix Taq (TaKaRa Biotechnology Co., Dalian, Liaoning Province, China), and 10.5 μL double distilled water (ddH2O). The PCR amplification parameters were as follows: 4 min at 94˚C; 35 cycles of 30 s at 94˚C, 30 s at a primer-specific annealing temperature, 30 s at 72˚C and a final extension step at 72˚C for 10 min. PCR products were then separated using capillary electrophoresis and genotyped using an ABI 3730 DNA analyzer with a GeneScan 500 LIZ size standard (Applied Biosystems, Beijing, China). Finally, we screened eight primers with polymorphic loci (Table 2). Polymorphism information content (PIC) values of markers were calculated using Cervus 3.0.7 [45].

## Genetic diversity analysis

We used the eight abovementioned polymorphic SSR markers to analyze genetic diversity among grazed and ungrazed samples. First, GeneMarker version 2.6.0 (SoftGenetics, State College, PA, USA) was used to perform peak identification and fragment sizing. Then, we applied GenAlEx 6.5 [46] to convert the data format and to calculate the genetic parameters, including the observed number of alleles (Na), effective number of alleles (Ne), expected heterozygosity (He) and observed heterozygosity (Ho), gene flow (Nm), and Shannon's information index

**Table 2. Characteristics of eight microsatellite loci for S. breviflora.**

| Gene ID | Primer sequence (5'→3') | Product size | Repeats | Ta(˚C)[a] | PIC[b] |
|---|---|---|---|---|---|
| CL14453.Contig1 | GAGGAAGCGTCGATCGTGAC | 107 | (CGG)7 | 61.5 | 0.47 |
| | TGTCCACTTTCTGCTCCACG | | | | |
| Unigene12747 | GAGCGATGCAACGATTATATAGG | 116 | (CGACGC)5 | 58.3 | 0.38 |
| | CATCGTGAAGTGATAAGAAGCCT | | | | |
| CL616.Contig34 | AATCAGCTCGTCGGTATTTGAT | 124 | (GT)6 | 53.0 | 0.33 |
| | AGAGGGGAAGAACGAAATATCTG | | | | |
| CL1837.Contig15 | GGGGAGTTGGACTTGGTAGTG | 150 | (GGA)5 | 62.0 | 0.22 |
| | CTTAACCTCCCTTCTCCACCTT | | | | |
| CL15080.Contig2 | ATCGTCAAACTCCACCTAATCAA | 140 | (TC)10 | 54.3 | 0.15 |
| | AGGCAATATTGGCAACTCACTC | | | | |
| CL966.Contig12 | CACTGGGTTCTCTTCGTCTCC | 237 | (GA)7gg(GA)7 | 52.5 | 0.35 |
| | TCTCCTCCTGCATCTTTCGTC | | | | |
| CL.5285.Contig5 | TATAGGCAGTGGGGAGACGA | 192 | (CTC)7 | 55.0 | 0.42 |
| | AGCTTCGAGGGATGAGGAGA | | | | |
| CL966.Contig15 | CACCGGATGCAAAGAAACCG | 221 | (GA)7gg(GA)7 | 52.5 | 0.14 |
| | CCTCCTGCATCTTTCGTCCTC | | | | |

[a] Annealing temperature.

[b] Polymorphism information content.

(I). R version 3.6.3 was used to perform Student's test to explore the relations of genetic parameters between grazed and ungrazed populations.

# Results

## *De novo* assembly and unigene annotation

After sequence assembly and redundancy removal, we obtained 111,018 unigenes totaling 125,503,270 bp with an average length of 1130 bp. The N50 and CG contents were 1852 bp and 49.38%, respectively, which demonstrated adequate assembly quality (Table 3). For the functional annotation, 88,164 unigenes (79.41%) matched records in seven databases. The quantity of annotated unigenes in each database was as follows: Nr, 79,283 (71.41%); Nt, 83,097 (74.85%); SwissProt, 57,827 (52.09%); KEGG, 63,044 (56.79%); COG, 37,880 (34.12%); InterPro, 51,305 (46.21%); and GO, 50,089 (45.12%) (Table 4).

**Table 3. Statistical summary of S. breviflora transcriptome data.**

| Assembly statistics | RNA-seq |
|---|---|
| Number of unigenes | 111,018 |
| Total length (bp) | 125,503,270 |
| Mean length (bp) [a] | 1130 |
| N50 (bp)[b] | 1852 |
| GC (%)[c] | 49.38 |

[a] Mean length of the assembled sequences.

[b] The length of the content or unigene corresponding to the sequence, which is added to 50% of the total assembled bases when the assembled sequences are sorted from short to long.

[c] GC content.

**Table 4. Summary of functional annotations of the assembled unigenes.**

| Database | Number of unigene hits | Percentage (%) |
|---|---|---|
| Nr | 79,283 | 71.41 |
| Nt | 83,097 | 74.85 |
| SwissProt | 57,827 | 52.09 |
| KEGG | 63,044 | 56.79 |
| COG | 37,880 | 34.12 |
| Interpro | 51,305 | 46.21 |
| GO | 50,089 | 45.12 |
| Overall | 88,164 | 79.41 |

Based on the NR annotation, we found species with genes homologous to *S. breviflora*. The results showed that 46.02% of unigene sequences matched *Brachypodium distachyon*, followed by other species (28.93%), *Hordeum vulgare subsp. Vulgare* (11.06%), *Oryza sativa Japonica* group (7.15%), and *Aegilops tauschii* (6.84%) (Fig 1).

We matched unigenes to the GO database to clarify the cellular function of the gene products. We found that 50,089 unigenes were matched to more than 50 functional items, which were divided into three main groups: biological process, cellular component, and molecular function (Fig 2). The largest subcategory of biological process was "metabolic process" (26,800), followed by "cellular process" (26,513). Of the cellular component, "cell" and "cell parts" showed the most matches with unigenes at 26,939 and 26,833, respectively. Among the molecular function category, "binding" and "catalytic activity" showed the most matches at 24,708 and 22,732, respectively.

With the aim of validating protein functions, the COG database was used to compare with unigenes. A total of 37,880 unigenes were annotated into 25 COG functional categories (Fig 3). The "general function prediction only" was the cluster with the most unigenes (10,786), followed by "transcription" (7184), "translation, ribosomal structure and biogenesis" (7048), "cell cycle control" (6488), and "function unknown" (6314), whereas only a few unigenes were assigned to "extracellular structures" and "nuclear structure".

KEGG annotation was performed to identify the metabolic pathway to which unigenes belonged, and 63,044 unigenes had matches in the KEGG database. These unigenes were assigned to 133 pathways.

## Differential expression and pathway analysis

Using the NOI approach, we identified 686 DEGs, including upregulated (304 DEGs) and downregulated (382 DEGs). Then, a similarity search of the DEGs in the KEGG database was

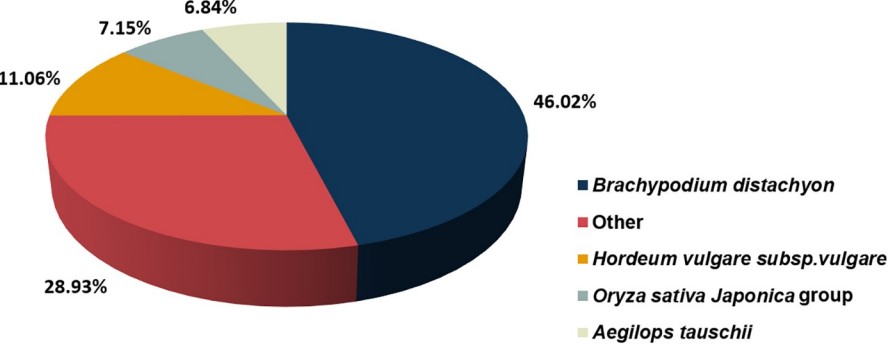

**Fig 1. Unigene homology searches against the NR database.**

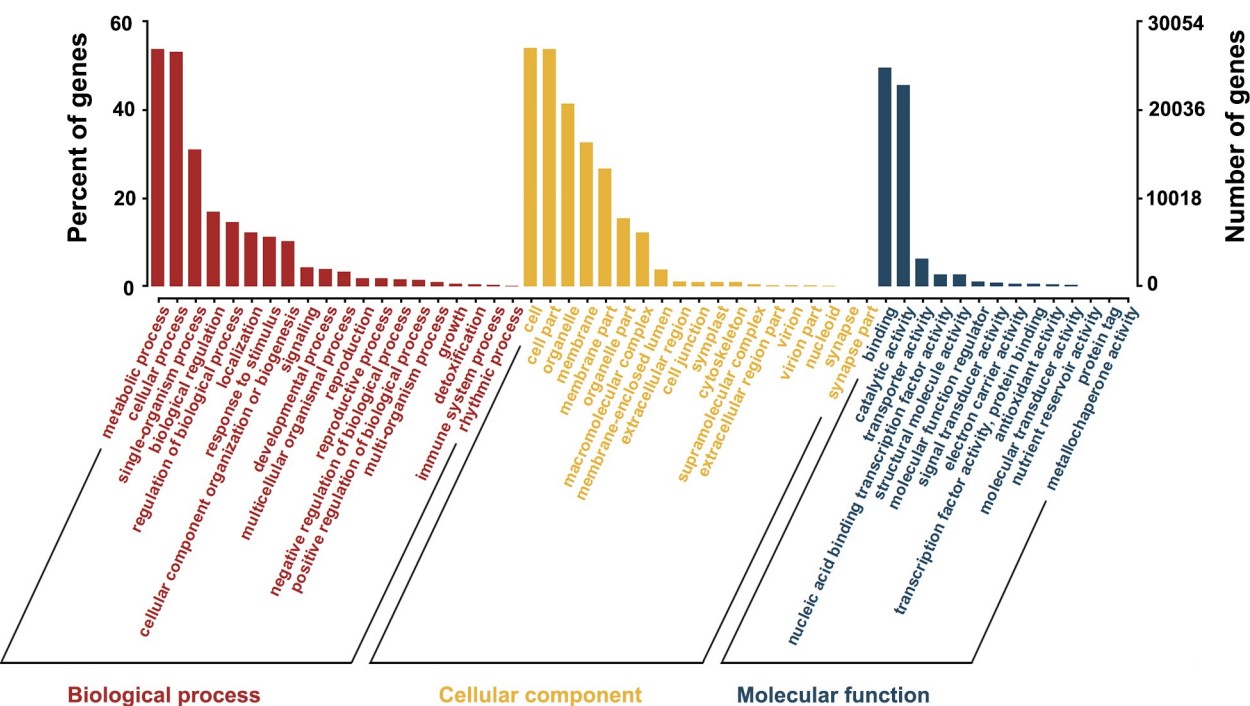

**Fig 2. GO classifications of the assembled unigenes.**

performed to explore the pathways associated with a significant enrichment in response to grazing. We found that these DEGs were sorted into 100 pathways. As presented in Table 5, the main enriched pathways included "alpha-linolenic acid metabolism" and "plant-pathogen interaction".

## Development and validation of grazing-related SSR markers

MISA was used to mine SSR markers from transcriptomes using the following criteria: mononucleotides with at least 12 repeats, di-nucleotides with 6 repeats, tri- and tetra-nucleotides with 5 repeats, and penta- and hexa-nucleotides with 4 repeats. We examined 111,018 sequences and identified 19,607 SSRs (S2 Table). Additionally, 11,963 tri-nucleotide SSRs and 386 quad-nucleotide SSRs accounted for 61.01% and 1.97% of all the SSRs, respectively, representing the most and the least of all SSR markers. The CCG/CGG trinucleotide repeats were the most abundant motifs detected in SSRs (5186, 26.45%), followed by the AG/CT (3117, 15.90%), AGG/CCT (2611, 13.32%), and AGC/CTG (1716, 8.75%) motifs (Table 6). The frequency of the remaining motifs accounted for 35.58% of the SSRs.

To develop grazing-related molecular markers, we preliminarily selected 21 DEGs and designed primers (S1 Table). After screening polymorphic SSRs, excluding primers with product sizes that did not meet the requirements and could not generate PCR products, eight grazing-related genes with SSR polymorphisms were found. In this study, PIC values ranged from 0.14 to 0.47, averaging 0.31, which indicated that these markers were informative and effective for genetic analysis (Table 2).

## Effects of grazing on genetic diversity

The eight primers yielded 29 alleles for 180 samples from the six sites. Among these primers, CL966.Contig12 and CL.5285.Contig5 produced the largest number of alleles, i.e. five alleles.

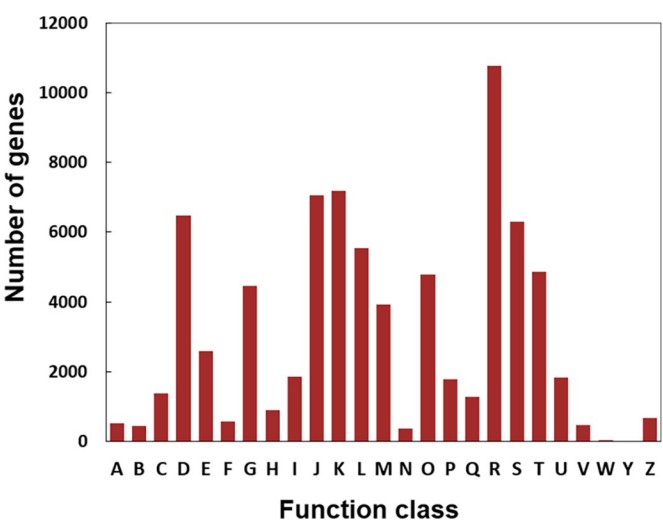

A: RNA processing and modification
B: Chromatin structure and dynamics
C: Energy production and conversion
D: Cell cycle control, cell division, chromosome partitioning
E: Amino acid transport and metabolism
F: Nucleotide transport and metabolism
G: Carbohydrate transport and metabolism
H: Coenzyme transport and metabolism
I: Lipid transport and metabolism
J: Translation, ribosomal structure and biogenesis
K: Transcription
L: Replication, recombination and repair
M: Cell wall/membrane/envelope biogenesis

N: Cell motility
O: Posttranslational modification, protein turnover, chaperones
P: Inorganic ion transport and metabolism
Q: Secondary metabolites biosynthesis, transport and catabolism
R: General function prediction only
S: Function unknown
T: Signal transduction mechanisms
U: Intracellular trafficking, secretion, and vesicular transport
V: Defense mechanisms
W: Extracellular structures
Y: Nuclear structure
Z: Cytoskeleton

**Fig 3. COG classifications of the assembled unigenes.**

The fewest numbers of alleles were exhibited in CL1837.Contig15 and Unigene12747 with two (Table 7). On average, there were four alleles per primer. Shannon's information index of the primers ranged from 0.31 to 0.92, with a mean of 0.62.

The genetic diversity of *S. breviflora* can be calculated using Shannon's information index (I) for each population. As Table 8 indicates, the grazed populations showed the highest Shannon's information index value, with a value of 0.64 (pop4 and pop6). In contrast, the ungrazed populations showed the lowest level of genetic diversity, with a value of 0.44 (pop2). The observed numbers of alleles (Na) ranged from 2.13 (pop2) to 3.00 (pop4), with an average of 2.71. The effective number of alleles (Ne) varied from 1.47 (pop2) to 1.71 (pop4), with an average of 1.62. Compared with the mean observed heterozygosity of 0.04 (Ho), the mean expected heterozygosity (He) value was high, with a value of 0.34. This result indicated that the inbreeding rate of *S. breviflora* is high, and the proportion of heterozygotes is relatively low. As shown in Table 9, both Shannon's information index (I) and expected heterozygosity (He) in grazed populations are higher significantly than that in ungrazed populations (p<0.05).

**Table 5. The top ten KEGG enriched pathway of DEGs.**

| KEGG terms | Gene number | Q value | Pathway ID |
|---|---|---|---|
| Alpha-linolenic acid metabolism | 13 | 2.90E-05*** | ko00592 |
| Plant-pathogen interaction | 57 | 9.96E-04*** | ko04626 |
| Pentose and glucuronate interconversions | 13 | 0.12 | ko00040 |
| Diterpenoid biosynthesis | 5 | 0.12 | ko00904 |
| Biosynthesis of secondary metabolites | 80 | 0.18 | ko01110 |
| Starch and sucrose metabolism | 20 | 0.23 | ko00500 |
| Linoleic acid metabolism | 4 | 0.27 | ko00591 |
| Phenylpropanoid biosynthesis | 25 | 0.61 | ko00940 |
| Sesquiterpenoid and triterpenoid biosynthesis | 3 | 0.61 | ko00909 |
| Glutathione metabolism | 7 | 0.61 | ko00480 |

***q<0.001, significantly enriched pathway of DEGs.

**Table 6. Distribution of simple sequence repeat types in *S. breviflora* transcriptome.**

| SSR motif | SSR motif numbers | Ratio (%) |
|---|---:|---:|
| Mono-nucleotide | 1059 | 5.40 |
| AC/GT | 678 | 3.46 |
| AG/CT | 3117 | 15.90 |
| AT/AT | 624 | 3.18 |
| CG/CG | 377 | 1.92 |
| Tri-nucleotide | 11,963 | 61.01 |
| AAC/GTT | 149 | 0.76 |
| AAG/CTT | 576 | 2.94 |
| AAT/ATT | 52 | 0.27 |
| ACC/GGT | 812 | 4.14 |
| ACG/CGT | 545 | 2.78 |
| ACT/AGT | 52 | 0.27 |
| AGC/CTG | 1716 | 8.75 |
| AGG/CCT | 2611 | 13.32 |
| ATC/ATG | 264 | 1.35 |
| CCG/CGG | 5186 | 26.45 |
| Quad–nucleotide | 386 | 1.97 |
| Penta–nucleotide | 542 | 2.76 |
| Hexa–nucleotide | 861 | 4.39 |

## Discussion

NGS strategies and their application to transcriptomics can be used to meet the increasing demands of acquiring accurate genome information. To clarify the transcriptome responses of *S. breviflora* to grazing, we compared the transcriptome changes under grazing and nongrazing conditions and found that 686 unigenes showed differential expression. These DEGs were significantly enriched in the "alpha-linolenic acid and metabolism" and "plant-pathogen interaction" pathways, which may enhance the adaptability of *S. breviflora* to grazing. Based on DEGs, we screened eight grazing-related SSR markers to explore the genetic diversity of *S. breviflora* and found that grazing has a positive effect on genetic diversity. Changes in gene expression levels help *S. breviflora* defend against grazing damage, and the increased genetic diversity provides the potential for its adaptation and evolution; all of these factors contribute to the survival of plants in different aspects.

### Grazing stimulates the expression of wound- and defense-related genes

As a main effect of grazing on herbage, wounding threat is inevitable and occurs over the entire life [35]. In addition to developing adaptive traits such as low heights, small leaf sizes, low root biomass [47], few vegetative tillers, and short internode lengths [48], plants also modulate the expression levels of related genes, for instance, genes related to the jasmonic acid (JA) family [49], to defend themselves against grazing damage. For *S. breviflora*, genes such as phospholipases $A_1$ (*PLA$_1$*) and 13-lipoxygenase (13-*LOX*), which are known to be involved in the synthesis of JA, were upregulated. For other plants, leaf wounding response were also related to JA synthesis [50, 51]. Singh et al. [52] elucidated the expression profile of rice PLA-encoding genes under salt, cold, and drought stress using microarray data and found that all the DEGs were upregulated. When tomato (*Solanum lycopersicum L.*) is exposed to methyl-jasmonate treatment and damaged, *SILOX4*, which belongs to the *13-LOX* subfamily, is highly expressed

**Table 7. Diversity statistics of the eight SSR markers across *S. breviflora* populations.**

| Locus | Na[a] | Ne[b] | Ho[c] | He[d] | I[e] | Nm[f] |
|---|---|---|---|---|---|---|
| CL14453.Contig1 | 4 | 2.23 | 0.05 | 0.55 | 0.92 | 6.76 |
| Unigene12747 | 2 | 2.00 | 0.02 | 0.50 | 0.69 | 4.28 |
| CL616.Contig34 | 4 | 1.63 | 0.02 | 0.39 | 0.67 | 4.16 |
| CL1837.Contig15 | 2 | 1.33 | 0.00 | 0.25 | 0.41 | 11.57 |
| CL15080.Contig2 | 3 | 1.19 | 0.05 | 0.16 | 0.34 | 14.99 |
| CL966.Contig12 | 5 | 1.64 | 0.00 | 0.39 | 0.73 | 11.52 |
| CL.5285.Contig5 | 5 | 1.90 | 0.13 | 0.47 | 0.88 | 6.26 |
| CL966.Contig15 | 4 | 1.17 | 0.00 | 0.14 | 0.31 | 7.11 |
| mean | 4 | 1.64 | 0.03 | 0.36 | 0.62 | 8.33 |

[a] Observed number of alleles.

[b] Effective number of alleles.

[c] Observed heterozygosity.

[d] Expected heterozygosity.

[e] Shannon's information index.

[f] Gene flow.

[53]. *PLA_1* and *13-LOX* are shared by grazing and other wounding response in different species, indicating that plants have some common mechanisms in defense response. Herbivore grazing, however, is a joint process that includes wounding, defoliation, and saliva deposition. Chen et al. [54] used rice to study grazing-induced pathway, and found the expression level of 1,3-beta-glucosidase increased. This gene was reported to be associated with resistance in plant, cell wall construction and modification [55, 56]. In our study, gene related to beta-glucosidase was upregulated under grazing, conferring resistance on *S. breviflora*.

Grazing provides opportunities for pathogens to infect plants [57]. When plants are infected by pathogens, they develop disease resistance by activating mitogen-activated protein kinase (MAPK) cascades [58]. Specifically, MAPK kinase kinases (MAPKKKs) phosphorylate MAPK kinases (MAPKKs), which then phosphorylate MAPKs [59]. In the present study, *MAPKK* genes showed upregulated expression under grazing, suggesting that *MAPKK* genes

**Table 8. Genetic diversity statistics of *S. breviflora* populations.**

| Code | Population size | Na[a] | Ne[b] | Ho[c] | He[d] | I[e] | PPB (%)[f] |
|---|---|---|---|---|---|---|---|
| 1 | 29 | 2.50 | 1.63 | 0.05 | 0.32 | 0.55 | 100.00 |
| 2 | 29 | 2.13 | 1.47 | 0.02 | 0.28 | 0.44 | 87.50 |
| 3 | 27 | 2.88 | 1.61 | 0.05 | 0.33 | 0.51 | 100.00 |
| 4 | 30 | 3.00 | 1.71 | 0.03 | 0.38 | 0.64 | 100.00 |
| 5 | 28 | 2.88 | 1.61 | 0.03 | 0.36 | 0.62 | 100.00 |
| 6 | 28 | 2.88 | 1.66 | 0.03 | 0.38 | 0.64 | 100.00 |
| mean | 29 | 2.71 | 1.62 | 0.04 | 0.34 | 0.57 | 97.92 |

[a] Na: Observed number of alleles.

[b] Ne: Effective number of alleles.

[c] Ho: Observed heterozygosity.

[d] He: Expected heterozygosity.

[e] I: Shannon's information index.

[f] PPB: Percentage of polymorphic bands.

**Table 9. Student's test of genetic parameters between ungrazed and grazed populations.**

|  | Na[a] | Ne[b] | Ho[c] | He[d] | I[e] |
|---|---|---|---|---|---|
| df | 4 | | | | |
| P value | 0.131 | 0.196 | 0.423 | 0.019* | 0.015* |

[a]Na: Observed number of alleles.

[b]Ne: Effective number of alleles.

[c]Ho: Observed heterozygosity.

[d]He: Expected heterozygosity.

[e]I: Shannon's information index.

*p<0.05.

contribute to the defense against invasion by microorganisms. Gene overexpression in the MAPK cascade defense mechanism occurs widely in plants, such as rice [60], *Arabidopsis* [61], cotton [62], and wheat [63].

## Grazing increases the genetic diversity of *S. breviflora*

Genetic diversity is a fundamental basis for plants to adapt to the environment, and to provide the potential for evolution. Genetic diversity is shaped by the balance between genetic drift, inbreeding, recombination, gene flow, mutation, and selection [64]. Grazing affects plant genetic diversity in different ways. Accompanied by trampling and livestock migration, grazing impacts genetic diversity of populations mainly through gene flow and genome-wide selection that changes the substitution rate of mutants [65].

Gene flow, one of the important forces leads to improving genetic diversity of natural plant populations, occurs by the spread of pollen, seeds, spores, and other genetic materials. *S. breviflora* is a facultatively selfing species. We found that pollen flow may promote sexual reproduction of *S. breviflora*. As Table 9 indicates, He in grazing populations are significantly higher than ungrazed populations. Therefore, pollen-mediated gene flow plays an important role in maintaining genetic diversity of *S. breviflora* populations while stressed by grazing [66]. In general, grazing affects plant reproduction by defoliation. *S. breviflora*, however, can complete life circle under heavy grazing by increasing reproductive investment, which brings about maintaining even increasing seed germination rate [67–69]. Grazing transfers seeds through livestock migration and fosters seedling establishment, resulting in the enhancement of seed-mediated gene flow, and consequently increasing genetic diversity [66], though seed-mediated gene flow was not mentioned in the present study. Another important force driven genetic diversity changes is gene polymorphism. So, we deliberately mined SSRs derived from DEGs of *S. breviflora*. As a result, we found a higher genetic diversity in grazing populations. This result suggests that grazing promotes the polymorphism of genes with relevance to grazing. These polymorphisms may endue *S. breviflora* with more grazing tolerance.

Other factors such as marker systems and grazing intensity might be taken in to account. SSR system is more efficient for genetic analysis than the ISSR system [70]. We observed that gene flow of SSR primers in *S. breviflora* was higher than that of ISSR primers in *S. grandis* [25] and *S. krylovii* [24], which varied from 4.16 to 14.99, with a mean value of 8.33 (Table 7). In addition, grazing intensity has also been taken into account in other studies. Shan et al. [25] indicated that the genetic diversity of *S. grandis* was the highest under light grazing. Similar results were suggested for *Elymus. nutans* in the study by Ma et al. [26]. For *S. krylovii* [24] and *S. purpurea* [71], the highest value appeared in moderately grazed populations. Contradictory results, including a negative relationship [72] and no correlation [73], were reported in other

studies. These contradictory results imply the complexity of grazing-induced genetic diversity that are affected by grazing intensity, differences in species, geographic locations and molecular marker systems.

## Conclusions

Overall, grazing activated plant defense-related genes and increased genetic diversity in *S. breviflora*. The desert steppe in Inner Mongolia has become severely degraded because of long-term overgrazing. The *S. breviflora* community, however, remains stable, and this species occur with *S. bungeana*, *S. klemenzii*, and *S. krylovii* in different types of steppes under diverse conditions of precipitation, temperature, and soil attributes. The reason for this may be explained in part by our findings, i.e., the enhancement of genetic diversity of *S. breviflora* under grazing improves the adaptability of the species to stressed environments. For maintaining genetic diversity of *S. breviflora*, a suitable grazing intensity is needed in order to implement rational use and conservation of desert steppe.

## Supporting information

**S1 Table. Twenty-one candidate microsatellite loci for *S. breviflora*.**
(DOCX)

**S2 Table. Frequency of classified repeat types.**
(XLSX)

## Author Contributions

**Conceptualization:** Jing Ren, Jianming Niu.

**Data curation:** Dongqing Yan, Jing Ren.

**Formal analysis:** Dongqing Yan.

**Funding acquisition:** Jianming Niu.

**Investigation:** Jing Ren, Yu Ding, Jianming Niu.

**Methodology:** Dongqing Yan, Jing Ren, Jiamei Liu, Yu Ding.

**Project administration:** Jianming Niu.

**Software:** Dongqing Yan, Jing Ren, Jiamei Liu, Yu Ding.

**Supervision:** Jianming Niu.

**Visualization:** Jiamei Liu.

**Writing – original draft:** Dongqing Yan.

**Writing – review & editing:** Jianming Niu.

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
