## [Decision Letter · Decision Letter 0]

8 Sep 2020

PONE-D-20-16248

De novo assembly, annotation, marker discovery, and genetic diversity of the Stipa breviflora Griseb. (Poaceae) response to grazing

PLOS ONE

Dear Dr. Niu,

Thank you for submitting your manuscript to PLOS ONE. After careful consideration, we feel that it has merit but does not fully meet PLOS ONE’s publication criteria as it currently stands. Therefore, we invite you to submit a revised version of the manuscript that addresses the points raised during the review process.

Several additional explanations and elaborations concurrent with the literature update are needed. The authors are advised to adapt the manuscript form, particularly Introduction and Discussion sections according to the suggestions stated in the reivewers' reports. Please be aware of the attachments with additional comments uploaded by R#2 and R#3.

We look forward to receiving your revised manuscript.

Kind regards,

Branislav T. Šiler, Ph.D.

Academic Editor

PLOS ONE

Journal Requirements:

Reviewers' comments:

Reviewer's Responses to Questions

**Comments to the Author**

1. Is the manuscript technically sound, and do the data support the conclusions?

Reviewer #1: Yes

Reviewer #2: Partly

Reviewer #3: Yes

2. Has the statistical analysis been performed appropriately and rigorously? 

Reviewer #1: Yes

Reviewer #2: Yes

Reviewer #3: Yes

3. Have the authors made all data underlying the findings in their manuscript fully available?

Reviewer #1: Yes

Reviewer #2: Yes

Reviewer #3: Yes

4. Is the manuscript presented in an intelligible fashion and written in standard English?

Reviewer #1: Yes

Reviewer #2: Yes

Reviewer #3: Yes

5. Review Comments to the Author

Reviewer #1: The work done on the grass species Stipa breviflora does not justify the need of transcriptomics for genetic diversity against grazing. Better to include the merits of the study for justification of NGS analysis. There are other well adapted techniques for studying genetic diversity. Also grazing is not a big problem for plant science research which limits the productivity or greening of the lands across the world. In addition, how the results will be helpful for further research programs.

Reviewer #2: The paper is a good study where the authors have tried to see changes in the gene expression due to grazing. However, the findings need to be re-discussed in light of following comments.

The authors have noted that genetic diversity has increased in grazed sites as compared to ungrazed sites and there is difference in gene expression (mainly wound healing and pathogen interaction groups), they have provided various hypothesis to their findings.

Grazing and biodiversity

How grazing increases biodiversity is not clear?? Grazing by animals can increase seed dispersal, only if grazing is continued beyond maturation stage. Usually grazing is done at vegetative stage and resulting in regeneration and delayed or no flowering, if grazing intensity is high. Whereas, if grazing is controlled and stopped before flowering or maturity, the seed set with smaller seeds will be observed as there will be little vegetative growth. Grazing animals as well as even birds, insects etc. help in seed dispersal, but it is not clear how they increase genetic diversity in a species which is self pollinating, highly inbreeding and have very low heterozyogisty. Authors should provide some explanation based on facts.

Grazing and up regulation of wound healing and plant pathogen interaction group of genes:

Is it effect of grazing by animals or simple defoliation or cut and carry system could have caused similar effects. There is certain phenomenon associated with the defoliation or cutting the plants manually/ mechanically or grazing, the growth hormones, wound healing genes, will be activated. Similarly in response to pathogen interaction which is expected to increase in association of wounds caused by animal grazing and movement, the defense mechanism will get activated and genes associated with these will be upregulated.

Similar studies of upregulation of such genes have been reported by other authors in different plants due to injury or wounding/ cutting/ defoliation and other stresses . How in the present study it is related specifically to grazing?? It can be a simple sounding carried out by any activity. The same sets of genes will again be at normal level when the wounding shock is overcome after some time. So it does not create new genetic diversity. It needs response from authors.

How DNA sequences have changed to resist grazing pressure?. Is it due to expression of genes or activities of few genes changing in response to stress

Sampling method : In perennial tussock forming species, 10 meter distance is too less for any genetic diversity.

Table 8 Except for population 2, not much difference is seen in rest 5 populations. Interpretation of data needs to look into this fact. Observed heterozygosity is more in ungrazed population 1 and 3 as compared to grazed population 4, 5, 6. Needs explanation.

Too broad assumption, supporting data needed.

The upregulation of genes are common response to cutting, defoliation or grazing. Does it mean change in genetic diversity? It is usual that genes get activated due to some injury or climatic shock but the genes are again at same level of regulation once that shock is overcome, It needs further clarification how it is enhancing genetic diversity??

How grazing increases biodiversity is not clear?? Grazing by animals can increase seed dispersal, only if grazing is continued beyond maturation stage. Usually grazing is done at vegetative stage and if grazing intensity is high it results in regeneration associated with delayed or no flowering. Animals can help in seed dispersal only if they are allowed to move in seed maturation stage.

The study would have been more informative if another aspect of manual defoliation been added and transcriptome analysis carried out in three groups, ungrazed, grazed and manually cut. The difference would have in that case highlighted the grazing demand.

Another aspects of three stage study, one ungrazed field, second during grazing or 5-10 days after grazing and third one month after stopping the grazing would have given clear picture whether these upregulation or down regulation of genes are stable or just transient phenomenon due to stress.

The authors should provide explanations on these points.

Reviewer #3: The revised manuscript “De novo assembly, annotation, marker discovery, and genetic diversity of the Stipa breviflora (Poaceae) response to grazing” is well-written, clear, and in my opinion definitely worth to be published. Figures and Tables are necessary and well designed.

This study, although it has rather methodological nature, presents interesting data on transcriptome response of Stipa breviflora on grazing. The authors developed eight polymorphic molecular markers and investigated genetic diversity of the taxon in grazed and ungrazed sites, and found relatively high level of S. breviflora genetic diversity occurred under grazing.

Remarks:

In my opinion the Discussion chapter is too long and over discussed, especially subchapter Grazing stimulates the expression of wound and defense-related genes, and in my opinion it should be definitely shortened.

The literature data should be supplemented on some basic and important position regarding taxonomy, phylogeny, genetic diversity, hybridisation and gene-flow in Stipa. There is almost completly neglected such literature data as:

Lv, X., He, Q. & Zhou, G. Contrasting responses of steppe Stipa ssp. to warming and precipitation variability. Ecol. Evol. 9, 9061–9075. https ://doi.org/10.1002/ece3.5452 (2019).

Nobis, M. et al. Hybridisation, introgression events and cryptic speciation in Stipa (Poaceae): a case study of the Stipa heptapotamica hybrid-complex. Perspect. Plant Ecol. Evol. Syst. 39, 125457. https ://doi.org/10.1016/j.ppees .2019.05.001 (2019).

Krawczyk et al. 2018. Plastid superbarcodes as a tool for species discrimination in feather grasses (Poaceae: Stipa). Sci. Rep. 8, 1924. https ://doi.org/10.1038/s4159 8-018-20399 -w.

Lu, S. L. & Wu, Z. L. On the geographical distribution of the genus Stipa L., China. Acta Phytotaxon. Sin. 34, 242–253 (1996).

Wu, Z. L. & Phillips, S. M. Stipa. In Flora of China (eds Wu, Z. Y. et al.) 196–203 (Science Press, Beijing, 2006).

Nobis et al. 2016. Stipa dickorei sp. nov. (Poaceae), three new records and a checklist of feather grasses of China. Phytotaxa 267(1): 29–39. http://dx.doi.org/10.11646/phytotaxa.267.1.3

Klichowska et al. 2018. Development and characterization of microsatellite markers for endangered species Stipa pennata (Poaceae) and their usefulness in intraspecific delimitation. Molecular Biology Reports 45(4): 639–643. https://doi.org/10.1007/s11033-018-4192-x

Klichowska et al. 2020. Different but valuable: Anthropogenic habitats as genetic diversityreservoirs for endangered dry grassland species – A case study of Stipa pennata. Ecological Indicators 111: 105998. https://doi.org/10.1016/j.ecolind.2019.105998

Baiakhmetov et al. 2020. Morphological and genome-wide evidence for natural hybridisation within the genus Stipa (Poaceae, sect. Leiostipa). Scientific Reports 10, 13803 https://doi.org/10.1038/s41598-020-70582-1

Nobis, et al. 2020. A revision of the genus Stipa (Poaceae) in Middle Asia, including a key to species identification, an annotated checklist and phytogeographic analysis. Ann. Mo. Bot. Gard. 105, 1–63. https ://doi.org/10.3417/20193 78

Nowak, et al. 2016. Vegetation of feather grass steppes in the western Pamir Alai Mountains (Tajikistan, Middle Asia). Phytocoenologia 46, 295–315. https ://doi.org/10.1127/phyto /2016/0145.

Line 443 '...S. breviflora is a self-pollinating species ' - This is only partially true. Stipa breviflora is not obligatory self-pollinator. Flowers of this species can be pollinated also by pollen of other individuals od this taxon, basides cross polination cases are also possible. That is why, there are many hybridisations and introgression events in Stipa, also with share of S. berviflora (see Baiakhmetov et al. 2020 Sci Rep. and Nobis et al. 2020, Annal Mis Bot Gard).

Line 484-486: '.... helpful for the restoration, conservation, and management of desert steppe' - This is very general conclusion which is suitable for abstract rather than for Conclusion of the paper. It would be much valuable if you could explained to readers how the study can help in conservation and restoration (having in mind that grazing impact on genetic diversity!).

The second question is that, whether grazing of steppe vegetation can be regarded as threat or rather not grazing should be regarded as such one. Steppes were always under such animal pression, so maybe even more important problem in this case could be impact of not grazing that could influence on less genetic diversity and probability of species extinction due to environmental or climatic.

Figure and table captions are unavailable or there are no such captions in the manuscript

All the other remarks to the manuscript the authors can find in the attached pdf version of the manuscript

6. PLOS authors have the option to publish the peer review history of their article (what does this mean?). If published, this will include your full peer review and any attached files.

Reviewer #1: No

Reviewer #2: **Yes: **Ajoy Kumar Roy

Reviewer #3: No

---

## [Author Response · Author response to Decision Letter 0]

1 Nov 2020

Thank you for your comments on our manuscript entitled “De novo assembly, annotation, marker discovery, and genetic diversity of the Stipa breviflora Griseb. (Poacea) response to grazing” (ID: PONE-D-20-16248). Those comments are very helpful for revising and improving our paper, as well as the important guiding significance to our further researches. We have studied the comments carefully and made corrections which we hope meet with approval. The main corrections are in the manuscript and the responds to the reviewers’ comments are as follows (the replies are in blue font). The page number and line number written in black are based on the “Revised Manuscript with Track Changes. docx” file. The page number and line number written in blue are based on the “Manuscript. docx” file. Because the PDF page number generated by the submission system is different from the original manuscript we uploaded, please refer to the line number to locate the modification.

---

## [Decision Letter · Decision Letter 1]

2 Dec 2020

PONE-D-20-16248R1

De novo assembly, annotation, marker discovery, and genetic diversity of the Stipa breviflora Griseb. (Poaceae) response to grazing

PLOS ONE

Dear Dr. Niu,

Thank you for submitting your manuscript to PLOS ONE. After careful consideration, we feel that it has merit but does not fully meet PLOS ONE’s publication criteria as it currently stands. Therefore, we invite you to submit a revised version of the manuscript that addresses the points raised during the review process.

Both reviewers suggested acceptance of the revised manuscript. However, some of the authors' responses are not conceived on scientific facts. I put remarks where the revision is still needed:

L91-92: "Compare with traditional..." - wrong tense.

L95-96: Expression "microcosmic interaction in organisms" is rather picturesque, but fails to refer to scientific facts.

L435-439: Sentences: "This result suggests that grazing promotes the polymorphism of genes with relevance to grazing." and "Under long-term grazing pressure, more individuals with grazing tolerance survive, resulting in the increase of genetic diversity." cannot be true, but just the opposite. If grazing tolerance has the genetic background, then narrowing variation of a phenetic character would result in increasing homozigosity of the genes involved in such a trait, which through generations leads to deletion of rare alleles, and finally to the decrease in genetic variations (of the genes under the selective pressure). Authors should further elaborate the presented fact and to bring their stance throughout the text.

L141-142: "principle of population genetics" does not describe how to gather the samples for a study. That may be "General Principles of Good Sampling Practice".

L384-385: "For other plants, leaf wounding response were also related to JA synthesis." - please provide references here.

L396: Increment of 1,3-beta-glucosidase expression uner the mechanical wounding should be further elaborated and referenced.

L313: "number of alleles with 5" - wrong formulation.

We look forward to receiving your revised manuscript.

Kind regards,

Branislav T. Šiler, Ph.D.

Academic Editor

PLOS ONE

Reviewers' comments:

Reviewer's Responses to Questions

**Comments to the Author**

1. If the authors have adequately addressed your comments raised in a previous round of review and you feel that this manuscript is now acceptable for publication, you may indicate that here to bypass the “Comments to the Author” section, enter your conflict of interest statement in the “Confidential to Editor” section, and submit your "Accept" recommendation.

Reviewer #2: All comments have been addressed

Reviewer #3: All comments have been addressed

2. Is the manuscript technically sound, and do the data support the conclusions?

Reviewer #2: Yes

Reviewer #3: Yes

3. Has the statistical analysis been performed appropriately and rigorously? 

Reviewer #2: Yes

Reviewer #3: Yes

4. Have the authors made all data underlying the findings in their manuscript fully available?

Reviewer #2: Yes

Reviewer #3: Yes

5. Is the manuscript presented in an intelligible fashion and written in standard English?

Reviewer #2: Yes

Reviewer #3: Yes

6. Review Comments to the Author

Reviewer #2: The paper can be accepted now. It is improved with addressing comments of all reviewers. Authors have provided proper response to all the comments.

Reviewer #3: I have no further comments to the submitted manuscript. All the answers on my remarks have been addressed, and in my opinion the ms should be accepted for publication in PlosOne journal just as it is now.

7. PLOS authors have the option to publish the peer review history of their article (what does this mean?). If published, this will include your full peer review and any attached files.

Reviewer #2: **Yes: **Ajoy Kumar Roy

Reviewer #3: No

---

## [Author Response · Author response to Decision Letter 1]

4 Dec 2020

We have studied editor's comments carefully and made revisions on the original manuscript. The corrections are in the manuscript and the response to the editor's comments are in the "Response to Reviewers.docx". Please let us know if you have any question regarding the manuscript.

---

## [Editor Report · Decision Letter 2]

7 Dec 2020

De novo assembly, annotation, marker discovery, and genetic diversity of the Stipa breviflora Griseb. (Poaceae) response to grazing

PONE-D-20-16248R2

Dear Dr. Niu,

We’re pleased to inform you that your manuscript has been judged scientifically suitable for publication and will be formally accepted for publication once it meets all outstanding technical requirements.

Kind regards,

Branislav T. Šiler, Ph.D.

Academic Editor

PLOS ONE
---

## [Editor Report · Acceptance letter]

10 Dec 2020

PONE-D-20-16248R2 

*De novo* assembly, annotation, marker discovery, and genetic diversity of the *Stipa breviflora* Griseb. (Poaceae) response to grazing 

Dear Dr. Niu:

I'm pleased to inform you that your manuscript has been deemed suitable for publication in PLOS ONE. Congratulations! Your manuscript is now with our production department. 

Kind regards, 

on behalf of

Dr. Branislav T. Šiler 

Academic Editor

PLOS ONE